# Controlling Neural Network Smoothness for Neural Algorithmic Reasoning

**David A. Klindt**                                                    *klindt.david@gmail.com*

**Reviewed on OpenReview:** *https://openreview.net/forum?id=JnsGy9uWtI*

## Abstract

The modelling framework of *neural algorithmic reasoning* (Veličković & Blundell, 2021) postulates that a continuous neural network may learn to emulate the discrete reasoning steps of a symbolic algorithm. We investigate the underlying hypothesis in the most simple conceivable scenario – the addition of real numbers. Our results show that two layer neural networks fail to learn the structure of the task, despite containing multiple solutions of the true function within their hypothesis class. Growing the network's width leads to highly complex error regions in the input space. Moreover, we find that the network fails to generalise with increasing severity i) in the training domain, ii) outside of the training domain but within its convex hull, and iii) outside the training domain's convex hull. This behaviour can be emulated with Gaussian process regressors that use radial basis function kernels of decreasing length scale. Classical results establish an equivalence between Gaussian processes and infinitely wide neural networks. We demonstrate a tight linkage between the scaling of a network weights' standard deviation and its effective length scale on a sinusoidal regression problem, suggesting simple modifications to control the length scale of the function learned by a neural network and, thus, its *smoothness*. This has important applications for the different generalisation scenarios suggested above, but it also suggests a partial remedy to the brittleness of neural network predictions as exposed by adversarial examples. We demonstrate the gains in adversarial robustness that our modification achieves on simple image classification problems. In conclusion, this work shows inherent problems of neural networks even for the simplest algorithmic tasks which, however, may be partially remedied through links to Gaussian processes.

## 1 Introduction

The two most prominent paradigms in artificial intelligence research are discrete, symbolic algorithms on the one side, and continuous, neural information processing systems on the other (Fodor & Pylyshyn, 1988; Natarajan, 1989; Marcus, 2003). While systems of the latter kind have caused a revolutionary transformation of the field, they are often plagued by hard challenges, such as robustness to changes in the input distributions, for which algorithmic approaches can provide worst-case performance guarantees. Crucially, we know that both approaches are deployed by humans, akin to Kahneman's 1 and 2 reasoning systems (Kahneman, 2011), and that, therefore, algorithms must be implemented in biological neural networks in the human brain (Zador et al., 2022). For instance, observing a scene in the world, we know that it is represented and processed in the distributed representation of neural activity in visual cortex. However, the same scene is also represented when we describe it with the use of symbols and the syntax of our language (Yildirim et al., 2020). Thus, one of the most mysterious questions in neuroscience as well as in artificial intelligence research is: *where and how do these two representation systems interact?*

The concept of *neural algorithmic reasoning* (Veličković & Blundell, 2021) is a recent proposal for a modeling framework at the intersection between symbol processing algorithms and continuous distributed information processing systems (see also Smolensky, 1990; Bear et al., 2020; Sabour et al., 2017). As an illustrative example of this hybrid approach, we can think of a robot equipped with a neural network that processes

input images to extract, e.g., its position in space (visual encoder); this is coupled with a planning algorithm (e.g., *Dijkstra*) that computes the shortest path from the current location to the target; that output is then fed into another neural network (motor controller) which translates the symbolic action plan into continuous motor control outputs. The obvious question is how such a hybrid architecture may be trained, since we usually require differentiability of the whole system for end-to-end training. To solve this, Veličković & Blundell (2021) propose training a neural network to approximate the output of the algorithm in the middle of the model. This differentiable approximation would then allowing gradient flow from motor controller to visual encoder for effective end-to-end training.

The purpose of our work is to investigate the feasibility of neural algorithmic reasoning in one of the most simple conceivable settings: the addition of real numbers. Integer calculus and floating-point arithmetic in binary (symbolic) representations have previously received more attention (Nogueira et al., 2021; Talmor et al., 2020; Jiang et al., 2019; Thawani et al., 2021; Zhou et al., 2022; Hendrycks et al., 2021; Bansal et al., 2022). By contrast, we want to know if a simple multilayer perceptron (MLP, (Rosenblatt, 1958)), i.e. the basic building block of most neural networks, can learn to add real-valued numbers on a compact domain such as the unit disc. Specifically, we look at two layer MLPs which are particularly interesting because: i) they contain a parameter subspace that perfectly solves the task (see below), ii) they can provide an approximation of unlimited precision in the infinite width limit (Hornik et al., 1989), iii) they reveal an interesting failure case that prompts further study of neural network functions.

In summary, in this paper we find that artificial neural networks are unable to learn the simple function of adding real numbers, even if abundant training data is available, leaving nonlinear, uneven regions of error within the training domain and struggling to extrapolate beyond. A comparison to Gaussian processes suggests a simple partial remedy, exploiting classic results (Neal, 1996) about the equivalence between these two model classes, based on a correct adjustment of the smoothness of the learned function. We show that these modifications also translate to increased adversarial robustness on handwritten character image recognition (Goodfellow et al., 2014). This has important implications for real world applications, such as self-driving car vision controllers that need to be robust to shifting input statistics, where smooth, generalisable model functions are required. It also aligns with the motivation behind neural algorithmic reasoning to build differentiable models that generalise as broadly as classical algorithms with worst case analytical performance guarantees. Finally, we are interested in this minimal setting where seeing how a neural network fails to learn the correct algorithm reveals an interesting phenomenon that advances our basic understanding of the functions learned by neural networks and how to control their smoothness.

## 2   Background

Simple mathematical reasoning (Saxton et al., 2019; Charton, 2021) and, specifically, the addition of real numbers is a particularly interesting setting because it is so simple, while still elucidating important functional complexity of neural networks (Hendrycks & Dietterich, 2019) (Fig. 1) and clearly exposing the difficult *inductive inference* from an infinite look-up table to the proper representation of an algorithm (Henderson, 2022). Moreover, we know that the hypothesis class of neural networks with rectified linear unit (ReLU) activation functions trivially contains a parameter subspace with the correct solution, i.e., given inputs $x \in \mathbb{R}^2$, the following set of functions

$$f(x) := W_2 \max(0, W_1 x + b_1) + b_2, \quad W_1 := \begin{pmatrix} \alpha & 0 \\ \beta & 0 \\ 0 & \gamma \\ 0 & \delta \end{pmatrix}, \quad W_2 := \begin{pmatrix} \alpha^{-1} \\ \beta^{-1} \\ \gamma^{-1} \\ \delta^{-1} \end{pmatrix} \tag{1}$$

with $\alpha, \gamma \in \mathbb{R}_{>0}$, $\beta, \delta \in \mathbb{R}_{<0}$ and $b_1, b_2 := \mathbf{0}$ are a subspace of the network parameters that correspond to a perfect representation of the desired output $y = x_1 + x_2$.

Given input and output training pairs, a neural network can learn to approximate this function. However, it is an open question how accurate the approximation will be within the compact domain of the training data; possibly in the limit of infinite data and a model that is a universal function approximator (Hornik et al., 1989). Secondly, it is unclear how the model will generalise outside the domain of the training data.

Ideally, the concept of addition should only require a limited amount of training examples to understand the underlying algorithm for producing the correct answer on any pair of inputs.

A crucial difference between algorithms and neural network solutions is the way they generalise to different inputs (Marcus, 2003; Veličković & Blundell, 2021; Zador et al., 2022). Addition is defined on all numbers in $\mathbb{R}$, but the approximation learned by a NN can only observe a subset of those inputs in its training data (i.e., *i.i.d.* – independent identically distributed). Recent discussions have investigated this from the point of view of interpolation versus extrapolation (Nakkiran et al., 2021; Schott et al., 2021). Overparameterised NN exhibit a *double descent* phenomenon, which is thought to improve their generalisation performance by interpolating between training points (Chatterji et al., 2021) – although see (Balestriero et al., 2021). Going beyond the convex hull of the training data would, by contrast, require the ability to extrapolate to a new domain (*o.o.d.* – out of distribution). This is one of the key motivations behind neural algorithmic reasoning and it has recently been explored with transformers (Nogueira et al., 2021; Kim et al., 2021; Anil et al., 2022; Zhou et al., 2022; Charton, 2021; Zhang et al., 2021) and recurrent neural networks Bansal et al. (2022); Linsley et al. (2018); Schwarzschild et al. (2021). Apart from extensions of the compact training data domain, we also study robustness to specific distribution shifts such as added Gaussian noise (Hendrycks & Dietterich, 2019; Rusak et al., 2020). We also study adversarial robustness (Goodfellow et al., 2014), which can be thought of as worst-case distribution shifts, this has recently been studied in two layer MLPs (Dohmatob & Bietti, 2022) and is specifically interesting as it relates to Lipschitz constants (Virmaux & Scaman, 2018) and a neural networks smoothness (see section 3.5).

An important caveat is that the point of this study is not to find a tailored solution to the problem of adding real numbers. There are many handcrafted fixes to the specific experimental setup in this paper, which i) reduce the size of the network down to the exact required dimensions (i.e., 4, see equation 1), ii) encourage sparsity to switch off all superfluous units, or iii) use more involved transformer-based models (Vaswani et al., 2017; Zhou et al., 2022) to solve the task. However, none of these settings are of interest for the present research question: i) and ii) provide solutions only to this specific problem without any transferable insights into neural network functions as we obtain in this paper (section 3.3); and iii) obscures the simple failure cases of MLPs, which are the minimal building blocks to study if we want to understand neural networks (Wang et al., 2020; Dohmatob & Bietti, 2022).

## 3 Results

### 3.1 Neural Networks and Gaussian Processes Learning Addition

We first investigate learning addition of real numbers in two dimensions. For this, we uniformly draw $D = 128$ points on the unit disc (see also Fig. 6)

$$\mathcal{D} := \{x \mid x \in \mathbb{R}^2, \ ||x||_2 \leq 1\}. \tag{2}$$

We train a two layer neural network with ReLU nonlinearities after the first layer to solve the addition task using a simple squared loss function (for additional experimental details see Appendix section A.1)

$$\mathcal{L}_{MSE}(f, x) = (f(x) - (x_1 + x_2))^2. \tag{3}$$

Across the top row of Fig. 1 we change the number of hidden units $N$ (i.e., the width) of the network and observe the effect on the learned solution. With few units ($N = 16$), the model exhibits the recently proposed polytope structure of neural network approximated functions (Balestriero et al., 2018). With more units we are entering the *interpolation regime* (i.e., zero training error), where recent work on double descent might suggest better performance (Nakkiran et al., 2021). While the performance does increase (Appendix A.2.1), we can see that the model uses the additional capacity to cut up the input space into increasingly complex, uneven regions. Interestingly, the network learns intricate, nonlinear ridges of good performance on which the training data lies. These ridges appear to be connected on continuous paths – an intriguing observation for future NN research. Importantly, these patterns suggest that larger ReLU networks learn sieve-like solutions of increasing complexity (see also Fig. 12), but they do not enter a qualitatively different

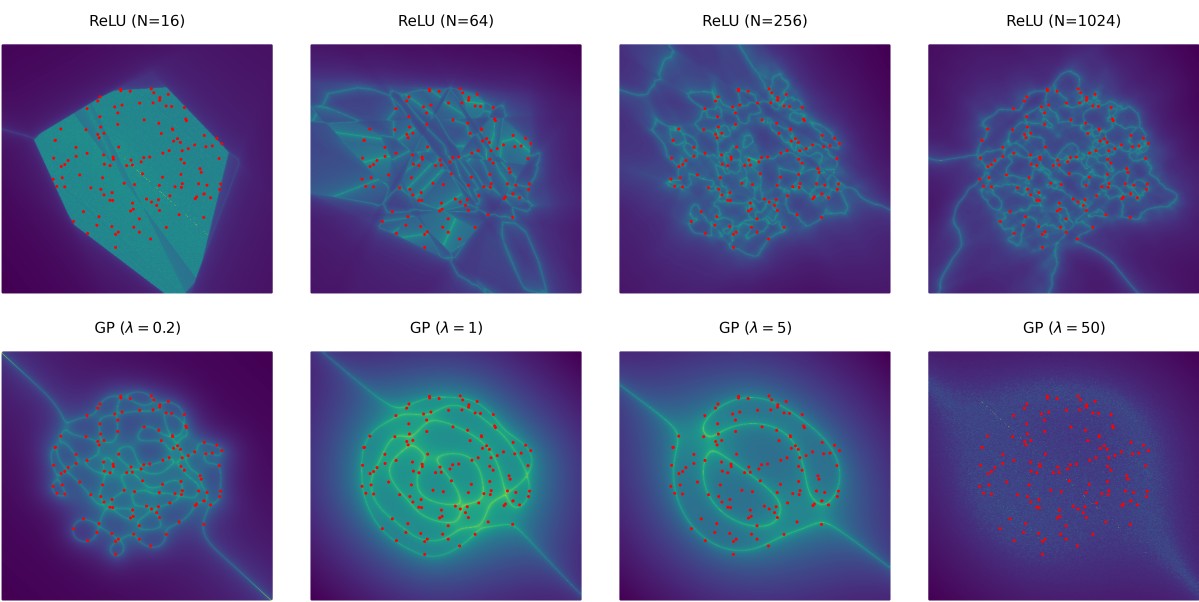

Figure 1: **Unit Disc Loss Surfaces**. These plots show the network error in input space (remember, we are trying to model $f : \mathbb{R}^2 \to \mathbb{R}, f(x) = x_1 + x_2, x \in \mathcal{D} \subseteq \mathbb{R}^2$), where brighter regions indicate lower error, i.e., colorscale shows the log of the loss function equation 3 (red dots – training data points). Top row shows the loss surface for a ReLU neural network with increasing number of hidden units ($N$, left to right). Bottom row shows the loss surface for a GP with a RBF kernel of increasing length scale ($\lambda$).

regime that would resemble the simple algorithmic solution to the task. We provide some initial exploration for why NNs fail to reach the optimal solution (equation 1) during optimisation in Appendix A.2.2.

We can produce a similar sequence of model behaviours with Gaussian process (GP) regressors with varying radial basis function (RBF) kernel scales. We use the standard GP implementation in `sklearn` (Pedregosa et al., 2011). Usually, this would include maximum likelihood length scale selection (optimum near $\lambda \sim 150$). However, setting this by hand lets us visualise the different solutions for *suboptimal* length scales. Specifically, we see that setting the length scale too low (Fig. 1 bottom left) forces the model to learn ridges of good solutions through the training data – similar to the NN model.[1]

Note that the anti-diagonal line passing through the origin indicates the null space of the target function. Precisely, adding any vector $n$ from the subspace $\mathcal{N} := \langle (1, -1) \rangle = \{a \cdot (1, -1) \mid a \in \mathbb{R}\}$ to an input $x$, does not change the output of the algorithm $y(x) = x_1 + x_2$, i.e.

$$y(x) = x_1 + x_2 = (x_1 + a) + (x_2 - a) = y(x + n), \quad \forall n = a \cdot (1, -1) \in \mathcal{N}. \tag{4}$$

Thus, on the anti-diagonal passing through the origin, the correct output is $y(x) = 0$ for all possible inputs $x = (0, 0) + \mathcal{N}$. We see that the chosen GP with a zero mean function, naturally, converges to this solution outside of the training data. For detailed average performance levels see Appendix A.2.1 and for density dependent solutions see Appendix A.2.6.

## 3.2 Out of Domain Generalisation

Building models that generalise outside of the training data is a key motivation behind neural algorithmic reasoning (Veličković & Blundell, 2021; Zhou et al., 2022). We adjust our setting slightly to study all aspects

---

[1]The geometry of these ridges can be studied analytically for GPs. We recognise that this might give meaningful insights into the appearance of similar ridges for NNs. However, we leave this open for future work.

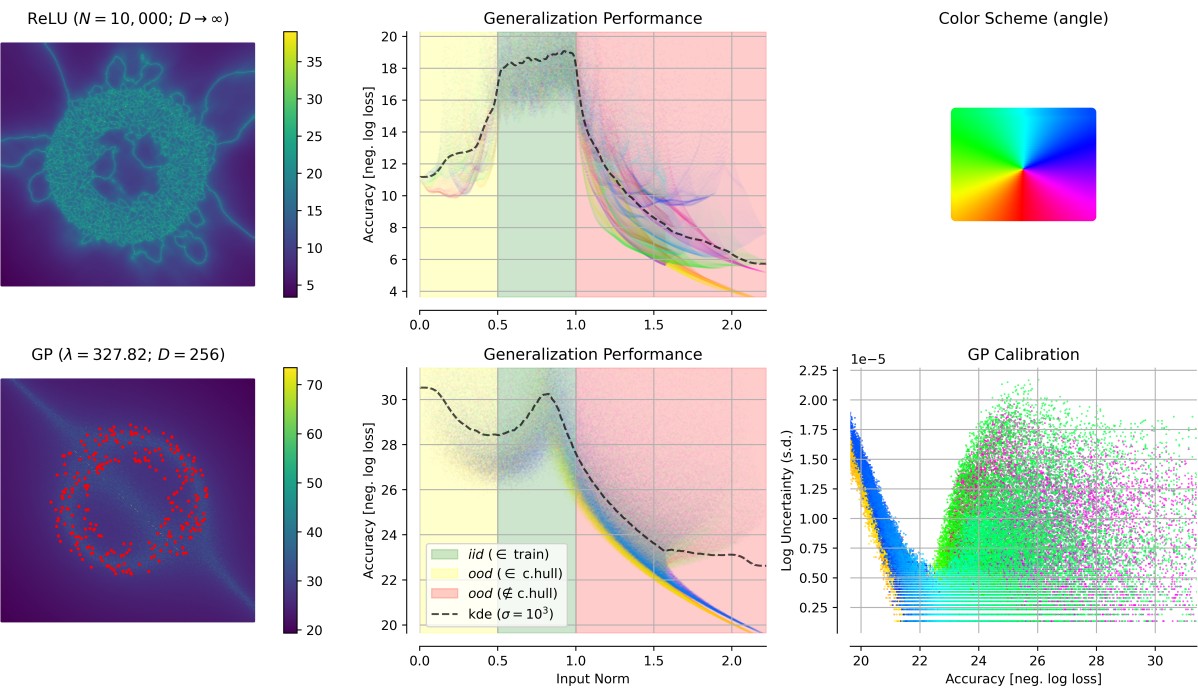

Figure 2: **Annulus Loss Surfaces and Generalisation Performance**. The first column shows the loss surfaces for a NN (top) with $N$ hidden units, and a nearly unlimited amount of training data; and (bottom) the loss surfaces for a GP with optimal length scale ($\lambda$) and $D = 256$ training data points, shown as red dots. The second column shows the negative log loss as a function of eccentricity (dots coloured by angle, black dotted lines radially averaged kernel density estimate – 'kde') in the three regimes of o.o.d. (within convex hull), i.i.d. and o.o.d. (outside of convex hull) generalisation for the NN (top) and GP (bottom). The third column (top) shows the colour scheme of the second column; and (bottom) the GP's uncertainty estimate.

of generalisation. Specifically, the training data is now randomly sampled from the annulus

$$\mathcal{A} := \{x \in \mathbb{R}^2 \mid 0.5 \leq \|x\|_2 \leq 1\}. \tag{5}$$

This allows us to test how well the model generalises:

- Within the training domain: $\mathcal{A}$ (Fig. 2, middle column, green).

- Outside of the training domain but within its convex hull: $\{x \in \mathbb{R}^2 \mid \|x\|_2 < 0.5\}$ (Fig. 2, yellow).

- Outside of the training domain and outside its convex hull: $\{x \in \mathbb{R}^2 \mid \|x\|_2 > 1\}$ (Fig. 2, red).

Moreover, to ensure that the findings from Fig. 1 do not depend on limited training data or finite network size, we set the number of hidden units to $N = 10,000$ and generate a new random batch for every gradient step (totalling $D = 256 \times 50,000 = 12,800,000$ training examples). As before, for full experimental details, please refer to Appendix section A.1.

The learned solution by the NN is shown in Fig. 2 top left. Again, we see an intricate, uneven pattern emerging within the training data domain (i.i.d.) with ridges of good performance but large valleys of bad predictions. We also observe that the model predictions deteriorate both inside the convex hull of the training data as well as outside. This is quantified in Fig. 2 top middle as a function of the input norm (we can also see the different ridges outside the training domain distinguished by their angles). For the GP, even with limited training data ($D = 256$), we see higher performance levels (Fig. 2 bottom middle) both on training data (green), in its convex hull (yellow) as well as outside (red). Moreover, for o.o.d. regions outside the

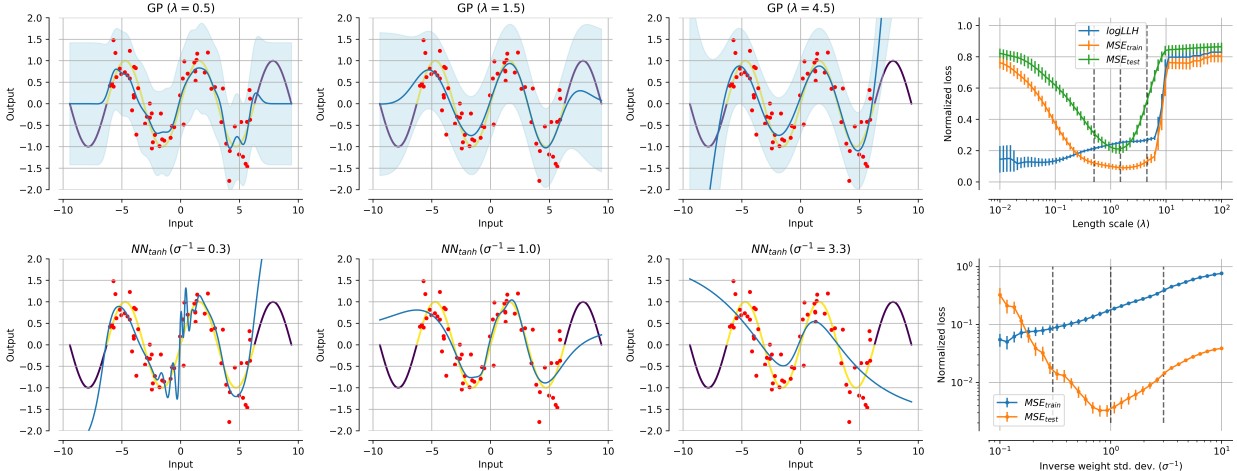

Figure 3: **Sinusoidal Regression**. Top row shows GP models with different length scales ($\lambda$) fitted to a sinusoidal regression problem with: true function (yellow i.i.d., purple o.o.d.), training points (red), model prediction (light blue), uncertainty (shaded, two standard deviations). The top right plot shows the different losses (mean squared error – MSE, and log likelihood) as a function of length scale (dotted lines indicate length scales in plots to the left). The bottom row gives the same regression plots for the NN with TanH activation and varying length scales ($\sigma^{-1}$) and (right) the training and test error as a function thereof.

training data's convex hull, where the performance starts dropping, the GP at least produces meaningful uncertainty estimates (Fig. 2 bottom right).

### 3.3 Setting the Length Scale of Neural Networks

The similarity between the solution patterns for neural networks with many hidden units (Fig. 1, top right) and GPs with short length scales (Fig. 1, bottom left) is not surprising given classical results (Neal, 1996). Briefly, Neal established that for NNs with a hyperbolic tangent (TanH) nonlinearity

$$\tanh(x) := \frac{\sinh(x)}{\cosh(x)} = \frac{e^{2x} - 1}{e^{2x} + 1} \tag{6}$$

and an infinite number of hidden units (with appropriate scaling of their initialisation variances) the distribution over learned functions (*ab initio*) becomes equivalent to that of a GP. This opens an interesting path forward in understanding and improving the neural network solution.

An important step in fitting a GP with a RBF kernel to data is finding the best length scale for the kernel. Effectively, this controls the smoothness (Lederer et al., 2019) of the resulting non-parametric model and, thus, the complexity of the hypothesis class (Vapnik, 1999). Translating this into the space of neural network functions is nontrivial in the general setting with ongoing work into the *Lipschitz* constants of trained models (Virmaux & Scaman, 2018; Fazlyab et al., 2019). In this section, thanks to the equivalence in the considered settings, we are able to investigate, in parallel, the length scale selection process both in GPs and NNs.

For easier visualisation of the learned input-output mapping we reduce complexity even further: The dataset is now simply a noisy sine function on $x \in [-2\pi, 2\pi]$ (for o.o.d., we extrapolate to $[-3\pi, 3\pi]$). Precisely, we are trying to model the function $y = \sin(x) + \epsilon$ with $\epsilon \sim \mathcal{N}(0, 0.25)$. In Fig. 3 we can see that there exists a sweet spot for the kernel's length scale (top middle), and that the model overfits (top left) for smaller length scales and underfits (top right) for larger length scales. This is confirmed quantitatively (top right) by looking at the likelihood (blue) as well as the training and test error as a function of the length scale.

To get a NN to behave like a GP, *we closely follow the construction in* (Neal, 1996) making the following changes to our NN model: i) We use a **TanH nonlinearity**; ii) We restrict the standard deviation (*std*)

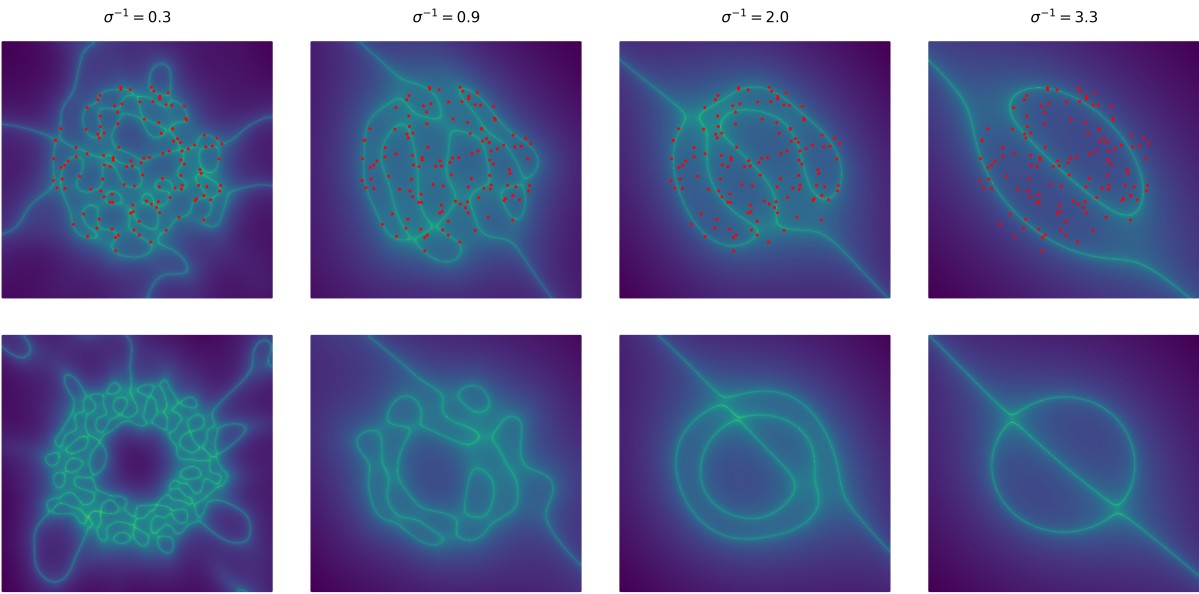

Figure 4: **Controlling the Smoothness of Neural Network Functions**. The four columns show learned TanH two layer NN loss surfaces for the two datasets: top, disc (section 3.1, $N = 10,000, D = 128$); bottom, annulus (section 3.2, $N = 10,000, D \to \infty$) for increasing length scales $\sigma^{-1}$ (left to right) producing an increasingly smooth mapping (for comparison, see also Neal (1996) Fig. 2.3).

of the weights to $\sigma\sqrt{N}$ where $N$ is the number of inputs to a layer and $\sigma$ a scaling factor. Importantly, in contrast to Neal (1996), we enforce this standard deviation (let $\mu(w) = \sum_i^N w_i$ denote the mean of the weights $w$) throughout training by using the scaled weights

$$\tilde{w} = w\frac{\sigma}{\sqrt{N}std(w)} = w\frac{\sigma}{\sum_i^N (w_i - \mu(w))^2}. \tag{7}$$

Intuitively, $\sigma^{-1}$ behaves like the length scale in GPs (Fig. 3 bottom), i.e., a smaller $\sigma^{-1}$ means larger weights and higher curvature/lower smoothness (potentially overfitting to the training data) whereas too large $\sigma^{-1}$ means very small weights and lower curvature/higher smoothness (potentially underfitting the training data) (see also Neal (1996) Fig. 2.3). Again, we establish the existence of an optimum length scale $\sigma^{-1}$ (Fig. 3, bottom right) that produces the smallest test error. In summary, these results suggest a simple remedy to improve NN approximations to simple algorithmic functions by making them more similar to GPs (with TanH nonlinearity) and correctly adjusting their length scale and, thus, their functional smoothness.

### 3.4 Controlling Neural Network Smoothness: Addition

Returning to the initial two examples of adding real numbers from a disc or an annulus, we can now observe the effect of varying the length scale ($\sigma^{-1}$) of a two layer NN with a large number of hidden units ($N = 10,000$, approaching the GP regime) and TanH activation functions (see Appendix A.1 for additional details).

On both datasets (Fig. 4) it is apparent that increasing the length scale makes the learned output function more smooth. Thus, the NN becomes qualitatively more similar to a GP with a well adjusted learning scale for its RBF kernel. We show in Appendix A.2.8 that the weight normalisation (equation 7) is crucial for these results, both in the case of addition and, for further verification and extension, multiplication.

### 3.5 Controlling Neural Network Smoothness: Classification

As further evidence in support of the claim that this approach effectively controls the smoothness of the learned NN input-output mapping, we turn towards the more complicated problem of performing image recognition on MNIST (LeCun et al., 1989) and CIFAR10 (Krizhevsky et al., 2009) under different distribution shifts. Briefly, we have a dataset of flattened images $x_i \in \mathbb{R}^{784}$ and labels $y_i \in \{1, ..., 10\}$. We are using the same two layer TanH model as above, with the minimal modification that the second layer output dimension is now 10 – i.e. the *logits*. All other details remain the same (see Appendix A.1), except for the objective function. Over training, we are minimising the standard cross-entropy loss between the output logits and the target label $\mathcal{L}_{CE}(f, x_i, y_i) = CE(f(x_i), y_i)$.

Since the input domain is now much more complex and high dimensional than in the previous settings, we have to adjust the different generalisation settings. Firstly, we look at additive white (Gaussian) noise of varying standard deviation (Fig. 5, right) – i.e. exploring random $L_2$ epsilon balls around the training data manifold (Hendrycks & Dietterich, 2019). Secondly, we look at worst case (adversarial) distribution shifts (Fig. 5, left) where we search for minimal perturbations that maximally change the output, which is a proxy for the model's smoothness (Goodfellow et al., 2014; Dohmatob & Bietti, 2022). More specifically, for a fixed maximum $L_2$ perturbation norm $c$ (horizontal axis in Fig. 5), we look for

$$\max_{\epsilon} \mathcal{L}_{CE}(f(x + \epsilon), y) \quad s.t. \quad ||\epsilon||_2 \leq c. \tag{8}$$

This second setting is particularly interesting: a) in high dimensions and complex input manifolds (here digit images) that defy simple separation of settings (Fig. 2, (Balestriero et al., 2021)), and b) because those

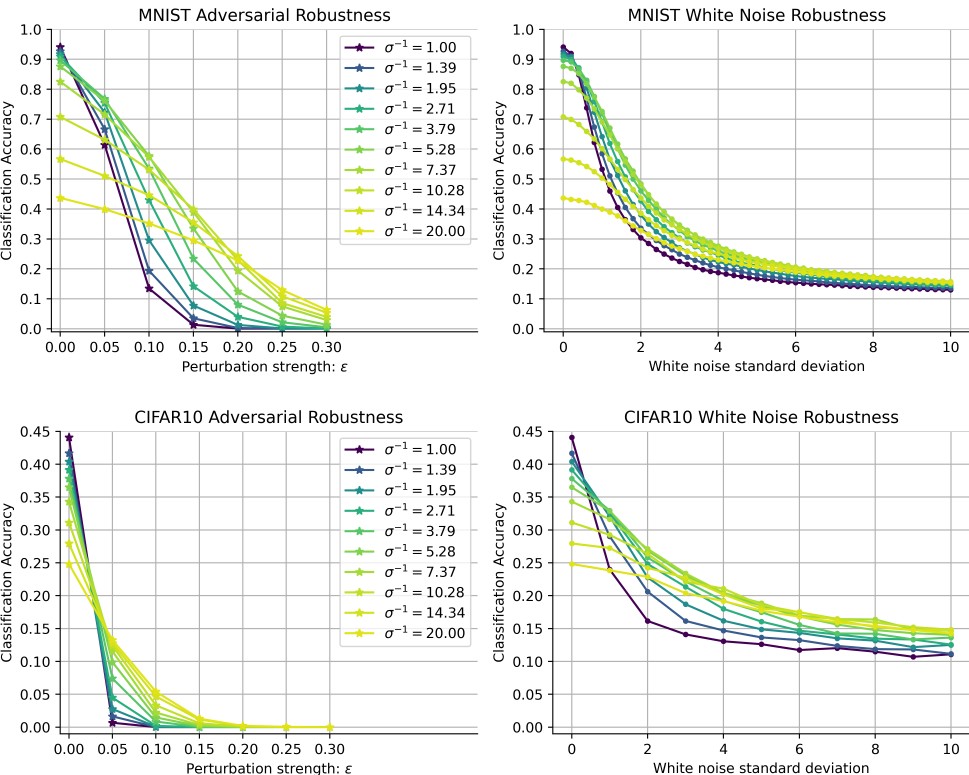

Figure 5: **Neural Network Classification Robustness**. Top, MNIST: Left) Accuracy as a function of (FGSM) perturbation strength for TanH NN models with different length scales ($\sigma^{-1}$); Right) Accuracy as a function of white noise standard deviation for the same models. Bottom, same for CIFAR10 dataset.

are the edge cases analysed when assessing the theoretical performance guarantees for classical algorithms (Veličković & Blundell, 2021).

Fig. 5 (left) shows that controlling the length scale of a neural network on handwritten character recognition does indeed increase the robustness to $L_2$ adversarial perturbations. Fig. 5 (right) shows that the robustness to white noise perturbations can be controlled in a similar way. In both cases, we can see a trade-off between the clean and the robust accuracy with large length scales increasing the robustness while decreasing the clean accuracy (Rusak et al., 2020). In a control experiment, we test whether the same robustness gains may be achieved with a simple control model using ReLU activations and standard weight decay (Appendix A.2.5). We find that this also increases adversarial robustness, however, to a smaller extent. To reduce computational costs, we have used the fast gradient sign (FGSM) attack (Goodfellow et al., 2014). However, in Appendix A.2.3 we show that we obtain the same qualitative results with the much stronger adaptive attacks implemented in `AutoAttack` (Croce & Hein, 2020).

## 4 Limitations

The proposed modifications are only applicable to simple two layer MLPs – i.e., the minimal example for which the complex input space behaviour (Fig. 1) arises (Wang et al., 2020). It is insightful to observe how the smoothness of this simple model class can be controlled in the GP regime (Lederer et al., 2019). However, this does not generalise to more layers (confirmed by preliminary experiments) and it is open for future work to investigate how the composite function of more than two layers contributes to global smoothness of a NN, potentially building on recent work extending the NN-GP equivalence to multilayer networks (Lee et al., 2017). To extend the current results we would need a closed form, or at least simple enough, expression for the length scale of a NN with more than two layers that allows us to control this parameter. Preliminary experiments on deeper models (Appendix A.2.4) reveal that gains in robustness do not transfer to multi-layer (convolutional) models in a straightforward way.

Even with the modifications, the network only becomes more similar to a tuned GP. However, there is still a gap between this smoother function approximation and the qualitative inference step of actually understanding the underlying arithmetic and learning the right solution for all possible inputs Henderson (2022). Put differently, the non-zero error rates, even for the GP, indicate that even smooth NNs are only taking us closer to the bold goal of learning how to infer and emulate simple algorithms.

While GPs show more desired behaviour on this task, again we want to highlight the fact that they do not offer a general solution to the problem that we are interested in, because they: i) fail to generalise properly to o.o.d. settings (see Figs. 1, 2); ii) scale poorly to problems with more data and larger input dimensions; and iii) cannot be integrated into a differentiable model pipeline, which is the original motivation for neural algorithmic reasoning. Moreover, using GPs instead of NNs (rather than as an inspiring analogy) would not allow us to extract the insights into neural network functions which we obtain in this paper (section 3.3). Therefore, we did not devote more time into studying different GP kernel and mean functions.

## 5 Discussion

The promise of neural algorithmic reasoning to combine distributed and discrete reasoning systems via differentiable NN approximations is intriguing (Veličković & Blundell, 2021). Here, we show that the functions learned by neural networks, even in one of the most simple conceivable examples of symbolic manipulation (real addition), is prone to learning a highly complex and uneven output mapping that falls short of learning the proper algorithmic target. There are many different perspectives on this problem, with previous approaches mostly focusing on transformers for manipulating arithmetic expressions (Thawani et al., 2021; Zhou et al., 2022; Anil et al., 2022; Nogueira et al., 2021; Jiang et al., 2019; Talmor et al., 2020). However, looking at the loss surface of the NN solution for a simple two layer MLP, reveals an intriguing structure of the loss surface and analogy to GPs that we decided to pursue in this work and that may also provide a useful pointer for future directions in basic NN theory. Specifically, as it challenges prior results stating that MLPs are able to perfectly learn linear functions (Xu et al., 2020) (under stricter theoretical assumptions) and that they are biased towards low frequency (i.e., smooth) solutions (Rahaman et al., 2019).

We have demonstrated how the NN mapping can be made more similar to that of a well-calibrated GP, specifically, by making it more smooth which also improved adversarial robustness. The smoothness is closely related to a neural networks' Lipschitz constant (Virmaux & Scaman, 2018). Whereas prior work derived GP Lipschitz constants depending on the specific kernel (Lederer et al., 2019), note that designing Lipschitz neural networks with (provably) bounded smoothness is an open research area (Fazlyab et al., 2019; Jordan et al., 2019). Also, the performance levels in Fig. 5 are far from state of the art robust models on this task, however, they eschew the need for (expensive) adversarial training (Kurakin et al., 2016; Wong et al., 2020) or generative inference (Schott et al., 2018). Thus, this is a proof-of-principle that the smoothness of a NN function can indeed be controlled with the construction proposed in this paper.

A different inductive bias (other than smoothness), is *symmetry* (Vapnik, 1999). For instance, the real numbers with the addition operation form an Abelian group and, thus, addition is commutative. Finding the right symmetries, such as permutation invariance of the inputs in the case of a commutative algorithm, is crucial in controlling the complexity of a learning problem (Bronstein et al., 2021). Intuitively, symmetries reduce the size of the search space by grouping parameter combinations that lead to the same function (cf., Entezari et al., 2021). We see in preliminary experiments that adding this additional constraint, does force the model to become permutation invariant (commutative) in its inputs (Appendix A.2.7), but it does not take us closer to properly learning the correct algorithm.

Here, we have closely followed the construction in Neal (1996) by normalising the network *weights*; future research might investigate the effect of normalising the network *activations* on the phenomena studied in this work. Prior findings suggests that batch normalisation (BN) (Ioffe & Szegedy, 2015), one of the most commonly used activation normalisation schemes (Ren et al., 2016), helps network training and generalisation by making the loss landscape smoother and reducing the norm of the weight gradient during optimisation (Santurkar et al., 2018). However, despite this increased smoothness of the loss landscape, BN does actually seem to hurt adversarial robustness (Benz et al., 2021; Singla et al., 2021) and make adversarial training more difficult (Wang et al., 2022; Walter et al., 2022). By contrast, test-time BN can help models deal with distribution shifts (Schneider et al., 2020), and normalisation in the form of feature competition (e.g., sparse coding, divisive normalisation) has been shown to increase adversarial robustness (Paiton et al., 2020).

Finally, while this study focuses on the smoothness of NN approximations to simple algorithms, the larger question remains still open how NNs can make the inferential step from any finite amount of data to an infinite look-up table (see MLST episode 061) – this amounts to Hume's *problem of induction* (Hume, 1896; Henderson, 2022) which may not be solvable with connectionist architectures (Fodor & Pylyshyn, 1988). Indeed, there remains a gap between the correct algorithm and its neural approximation which future research may seek to close, for instance, by taking a closer look at the biological solution (Zador et al., 2022). Surely, humans are a proof-of-concept that noisy and distributed processing systems (i.e., brains) can implement discrete symbolic algorithms (Fias et al., 2021). We hope that future research in this area will benefit more from interdisciplinary approaches that take inspiration across fields.

## Broader Impact Statement

As neural network applications become more and more ubiquitous in transforming data, for instance, in self-driving cars or autonomous flying drones, the need for reliable system grows proportionally. By studying the smoothness of neural networks, we hope to contribute to these efforts. Specifically, algorithms in high stake scenarios (such as traffic control) undergo thorough theoretical evaluation to ensure that they reach minimal performance targets even for worst case inputs. Analogously, controlling the smoothness of neural networks to equip them with this crucial aspect (in the spirit of neural algorithmic reasoning), takes small steps towards the same desideratum.

## Acknowledgements

The present paper has been shaped by insightful feedback from many people, including: Lukas Schott; Jing Yang Zhou, Eero Simoncelli and members of his group; Anthony Zador, Tatiana Engel, Alexei Koulakov and the Neuro-AI community at Cold Spring Harbor Laboratory; as well as the anonymous reviewers who

have suggested a number of additional experiments and valuable clarifications. We would like to thank all of these people for their time, thought and consideration which have helped to shape this paper into its final form, and which have, hopefully, provided them with some inspiration for their future scientific journeys.

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

# A Appendix

## A.1 Experimental Details

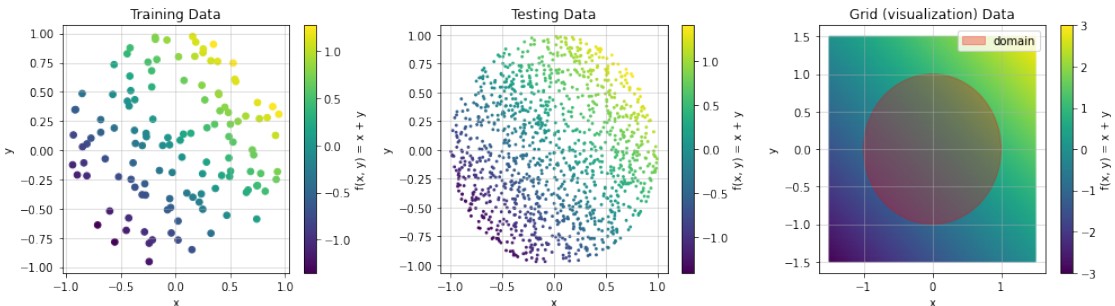

Figure 6: **Visualisation of Addition on Disc Datasets**.

To sample uniformly from the disc $\mathcal{D} := \{x \mid x \in \mathbb{R}^2, \ ||x||_2 \le 1\}$, we first sample a random angle $\theta \sim \mathcal{U}(0, 2\pi)$ and then a random length $r' \sim \mathcal{U}(0, 1)$ of which we take the square root $r = \sqrt{r'}$ to get a uniform sampling on the disc. These polar coordinates are then transform into standard euclidean coordinates $x_1 = r \cos \theta$ and $x_2 = r \sin \theta$ (see Fig. 6). For the Annulus $\mathcal{A} := \{x \in \mathbb{R}^2 \mid 0.5 \le ||x||_2 \le 1\}$, we simply sample the length from $r' \sim \mathcal{U}(0.25, 1)$.

We train for $50,000$ steps with the Adam optimizer (Kingma & Ba, 2014), an initial learning rate of 0.001 and a learning rate decay of 0.9. We verified in initial experiments that these settings led to best held-out performance and convergence of gradient descent for the tested models. For the adversarial robustness experiments in section 3.5, we use the fast gradient sign attack (Goodfellow et al., 2014).

## A.2 Additional Experiments

### A.2.1 Performance on Disc Addition

| (Model) | $\text{MSE}_{train}$ | $\text{MSE}_{test}$ | $\text{MSE}_{o.o.d.}$ |
|---|---|---|---|
| ReLU (N=16) | 2.93e-11 (2.39e-11 ) | 8.46e-06 (2.89e-06 ) | 1.99e-02 (4.49e-03 ) |
| ReLU (N=64) | 8.05e-08 (1.20e-08 ) | 6.22e-06 (1.10e-06 ) | 6.94e-03 (1.91e-03 ) |
| ReLU (N=256) | 1.46e-09 (3.20e-10 ) | 6.10e-06 (5.94e-07 ) | 2.46e-03 (4.31e-04 ) |
| ReLU (N=1024) | 1.95e-10 (6.30e-11 ) | 5.54e-06 (6.10e-07 ) | 1.48e-03 (1.51e-04 ) |
| GP ($\lambda$=0.2) | 2.09e-14 (1.79e-14 ) | 1.58e-03 (5.11e-04 ) | 1.55e-00 (1.12e-02 ) |
| GP ($\lambda$=1) | 3.67e-12 (1.09e-13 ) | 1.12e-10 (3.41e-11 ) | 9.29e-04 (8.08e-05 ) |
| GP ($\lambda$=5) | 1.18e-12 (4.50e-14 ) | 2.12e-12 (2.71e-13 ) | 4.24e-08 (3.77e-09 ) |
| GP ($\lambda$=50) | 2.43e-14 (2.69e-15 ) | 3.04e-14 (4.34e-15 ) | 6.97e-12 (9.52e-13 ) |

Table 1: **Performance on Disc Addition**. For ReLU networks with different numbers of hidden units ($N$) and GPs with different length scales ($\lambda$) corresponding to the models in Fig. 1. Reported is the mean squared error (MSE) on the training and test sets, as well as out of distribution $X_{o.o.d.} := [-1.5, 1.5]^2 \setminus \mathcal{D}$.

### A.2.2 Optimal Solution and Interpolation Experiments

Figure 7: **Interpolation Experiments**. Test loss as a function of the interpolation coefficient ($\alpha$) between models (see text) at initialisation ($\theta_0$), after training ($\theta_{sgd}$) and with the optimal solution ($\theta_{opt}$, equation 1). Top (bottom) row shows the value (index) of $\alpha$ on the horizontal axis for clearer visualisation. OLS (ordinary least squares) denotes models where the second layer weights ($W_2$) were set to the least squares solution on the training data set. Light colours indicate 10 random seeds and darker colours averages.

In this section, we provide a preliminary exploration for why NNs fail to find the optimal solution (equation 1). We primarily rely on one dimensional cross sections of the loss surface. Specifically, we look at interpolations from initialisation to trained models (Lucas et al., 2021; Vlaar & Frankle, 2022) as well as between trained models (Entezari et al., 2021; Ainsworth et al., 2022; Jordan et al., 2019). We choose the same settings as in Fig. 1 (top row, third column), i.e., $D = 128$ data points from the disc $\mathcal{D}$, $N = 256$ hidden units, standard two layer MLP with ReLU. As before, we train the network with the same settings (see section A.1) and denote the learned weights $\theta_{sgd}$. Further, we denote the weights at initialisation $\theta_0$ and the optimal solution $\theta_{opt}$ (equation 1). We repeat this experiment for 10 different random seeds.

First, we investigate whether interpolating from the network initialisation ($\theta_0$) to a solution produces a monotonically decreasing test loss (Lucas et al., 2021). In Fig. 7 (left column), we see that this is the case both when interpolating to the learned $\theta_{sgd}$ as well as the optimal $\theta_{opt}$ solution. Note that the loss curve of the former ($\theta_0$ to $\theta_{sgd}$) is lower for most of the interpolation distance, which might explain why a (greedy) gradient descent algorithm would prefer that direction. However, the interpolating to the optimal solution ($\theta_0$ to $\theta_{opt}$) ultimately achieves a much lower test error (up to machine precision), as expected.

Second, an alternative explanation for the failure to find the optimal solution is that as the number of units grows ($N \to \infty$), the (first layer) weights are virtually fixed and the model is forced to utilise the feature space provided at initialisation (cf. neural tangent kernel, (Jacot et al., 2018)). Consequently, during gradient descent, the model cannot reach the optimal solution (equation 1). To investigate this, we compute the ordinary least squares (OLS) solution for the second layer weights ($W_2$). Precisely, given the hidden activations $H \in \mathbb{R}^{N \times D}$ as $H_i = \max(0, W_1 X_i + b_1)$ for the training set ($X \in \mathbb{R}^{D \times 2}, Y \in \mathbb{R}^D$) we compute

$$\beta := (H^T H)^+ H^T Y \tag{9}$$

where $(H^T H)^+$ is the (Moore-Penrose) pseudo-inverse matrix of $H^T H$. We then use this as second layer weights $W_2 = \beta$ (with bias set to $b_2 = 0$) and test the model performance on the test set.

In Fig. 7 (middle column) we see that interpolating from $\theta_0$ to $\theta_{sgd}$ but computing the OLS solution for the second layer for each interpolation value ($\alpha$), we obtain a nearly constant loss curve. This suggests, that over training the quality of the first layer feature space (or equivalently, the quality of the model *modulo* optimal/OLS second layer weights) does not change and that most increase in performance during training comes from solving the (convex) optimisation of the second layer. By contrast, interpolating (*modulo* OLS) to the optimal solution ($\theta_{opt}$) stays approximately constant until, at the very end of the interpolation path, there is a sharp decrease in test loss (i.e., increase in the quality of the first layer feature space). This experiment suggests that: i) most training progress can be recovered by optimising the second layer during learning, which alone cannot achieve the optimal solution; and that ii) there is a sharp transition in model quality close to the optimal solution, until then the loss (*modulo* OLS) is almost constant.

Third, we may also ask: how does the loss surface behave for an interpolation between solutions, i.e., from $\theta_{sgd}$ to $\theta_{opt}$ (Entezari et al., 2021; Ainsworth et al., 2022; Jordan et al., 2019)? We see in Fig. 7 (right column, blue) that there is a *loss barrier* (Vlaar & Frankle, 2022), i.e., the loss increases from $\theta_{sgd}$ to $\theta_{opt}$. Recent work (Entezari et al., 2021) suggested that this barrier may be removed if permutations between network units are taken into account. In our simple model we can easily test this, again, by computing the OLS solution for the second layer which will account for any permutation $P \in \mathbb{S}_N$, since this can easily be absorbed into the OLS $P\beta = \tilde{\beta}$ estimate. We see in Fig. 7 (right column, green) that the loss is now approximately constant along the interpolation path again almost up to $\theta_{opt}$.[2]

In summary, these interpolation experiments suggest that: i) Interpolating to the trained solution leads to a faster (but ultimately smaller) decrease in loss (Fig. 7 left); ii) Most training progress can be recovered by simply optimising the second layer weights (Fig. 7 middle); iii) The quality of the first layer features (i.e., the loss *modulo* OLS) is nearly constant on the triangle ($\theta_0$, $\theta_{sgd}$, $\theta_{opt}$), but with a sharp transition close to $\theta_{opt}$ (Fig. 7 middle, right). Thus, it seems that the model does not find the optimal solution during optimisation because it stays in a regime where the quality of the first layer feature space is nearly constant and where all training progress can be reduced to solving the (convex) optimisation of the second layer weights. There is only a very sharp and narrow peak where the first layer feature quality rapidly increases around the optimal solution. Reaching this point in a vast surrounding space of constant first layer quality may be rather difficult with gradient descent optimisation.

---

[2]This supports the idea of *linear mode connectivity* modulo permutations (Ainsworth et al., 2022). Later work (Jordan et al., 2022) showed that rescaling individual units to maintain variance can also be important for linear mode connectivity; again, this linear transformation can be subsumed into the OLS $\beta$ weights.

### A.2.3 Stronger Adversarial Attacks

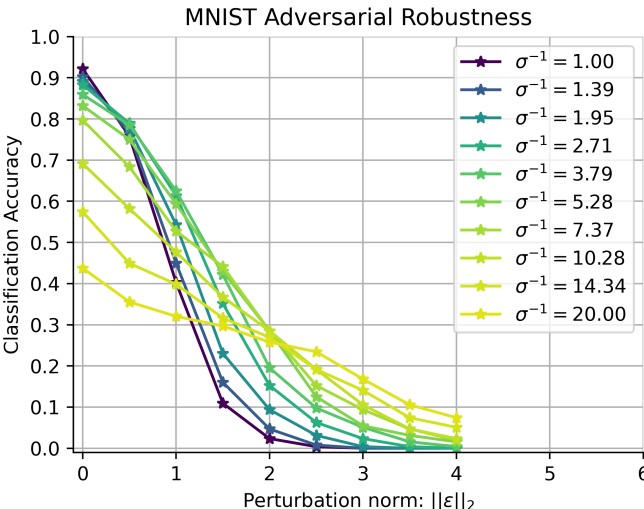

Figure 8: **Neural Network Classification Robustness**. Accuracy on MNIST as a function of ($L_2$) adversarial perturbation size ($||\epsilon||_2$) for TanH NN models with different levels of weight decay. Attacks computed with the stronger adaptive attacks implemented in `AutoAttack` (Croce & Hein, 2020).

### A.2.4 Multilayer Model for CIFAR10 Classification

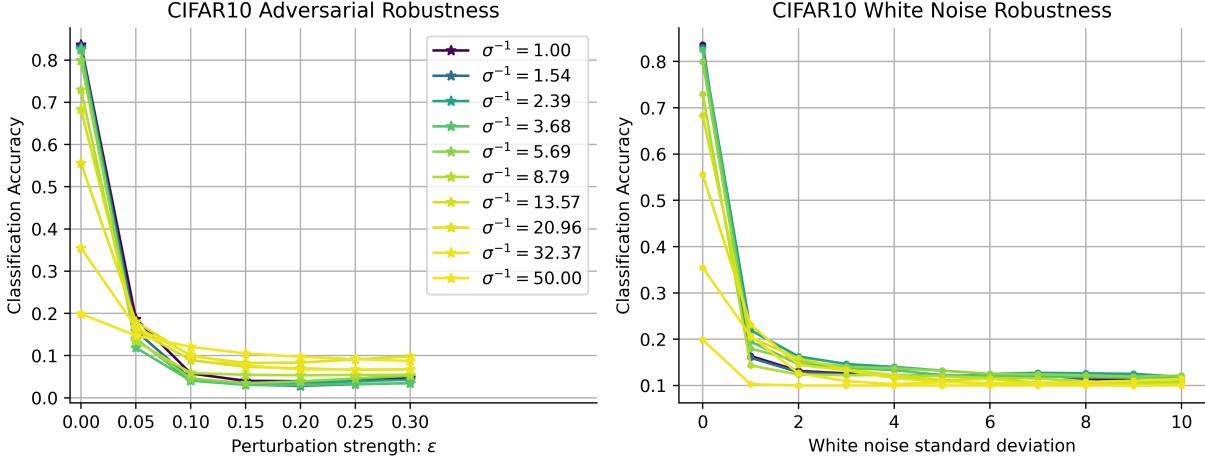

Figure 9: **CIFAR10 Classification with ResNet Models**. Accuracy on CIFAR10 as a function of (FGSM) perturbation strength for models with different length scales ($\sigma^{-}1$); Right) Accuracy as a function of white noise standard deviation for the same models.

We are using a standard `ResNet9` (He et al., 2016) implementation based on this public repository (kaggle). Importantly, we exchange every fully connected layer with the construction proposed in this paper (section 3.3), i.e., normalising the weights and using TanH nonlinearities. Secondly, we apply the same normalisation to every input channel of convolutional layers and also change their activation functions to TanH nonlinearities. We are using the same $\sigma$ for all layers. Notice, that these are specific design choices and that we leave it open to future research to explore different normalisation strategies or layer-dependent smoothness parameters $\sigma$ that could, potentially, improve the present results on deeper (vision) models.

## A.2.5 Weight Decay Robustness Comparison

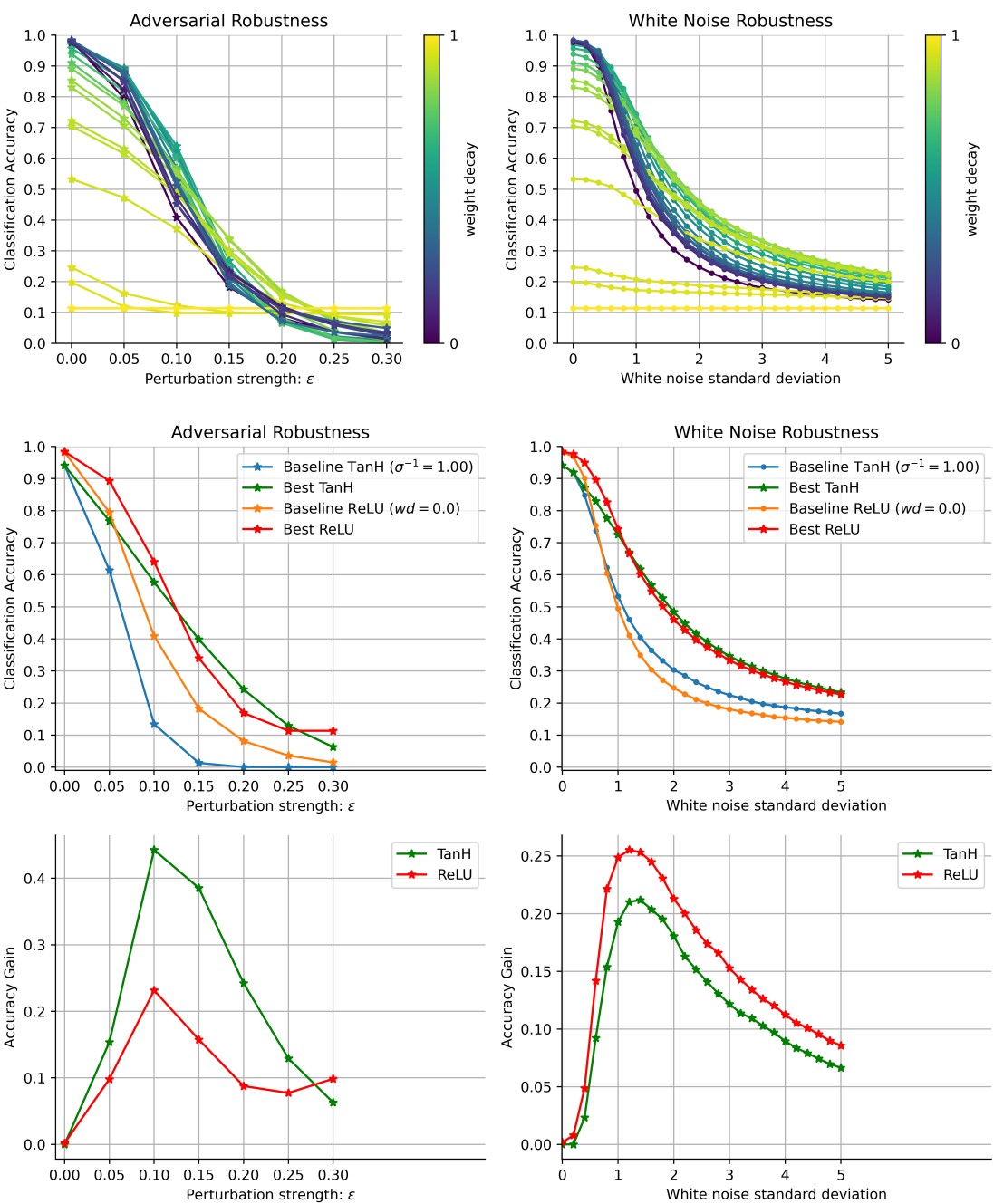

Figure 10: **Neural Network Classification Robustness**. Top left) Accuracy on MNIST as a function of (FGSM) perturbation strength for ReLU NN models with different levels of weight decay ($wd$). Top right) Accuracy on MNIST as a function of white noise standard deviation for the same models. Middle) Accuracy for baseline TanH ($\sigma = 1.00$) and ReLU ($wd = 0.0$) models for adversarial (left) and noise (right) perturbations, and best models, i.e., maximum performance over $\sigma$ ($wd$) for every perturbation size. Bottom) *Accuracy Gain*, i.e., difference between baseline and best (middle) for both TanH and ReLU models.

### A.2.6 Density Dependent Fitting

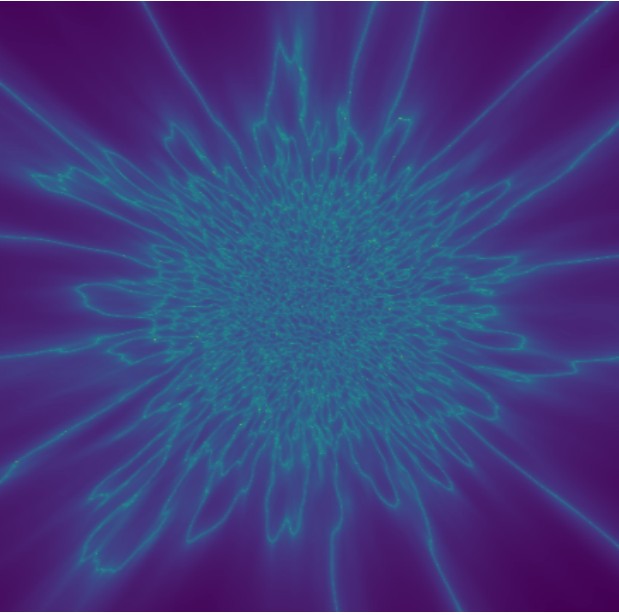

Figure 11: **TanH Neural Network Learning Addition on Gaussian Data**. Data drawn from standard normal distribution $\mathcal{N}(0,1)$. Colorbar indicates the log of the loss function (equation 3).

### A.2.7 Inductive Bias – Commutative Symmetry

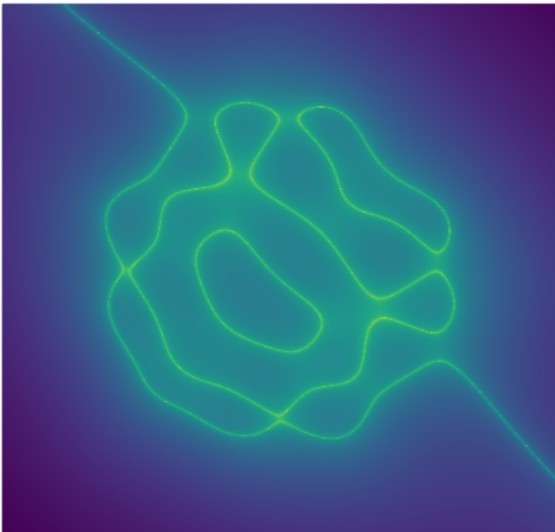

Figure 12: **Exploring Commutative Symmetry as Inductive Bias**. TanH model ($\sigma^{-1} = 1.0$) trained on disc addition task (Fig. 1), however, forcing commutativity in the outputs by changing the network function $f$ to $f_{sym}(x) = (f(x_1, x_2) + f(x_2, x_1))/2$. Colorbar indicates the log of the loss function (equation 3).

## A.2.8 Results on Addition and Multiplication

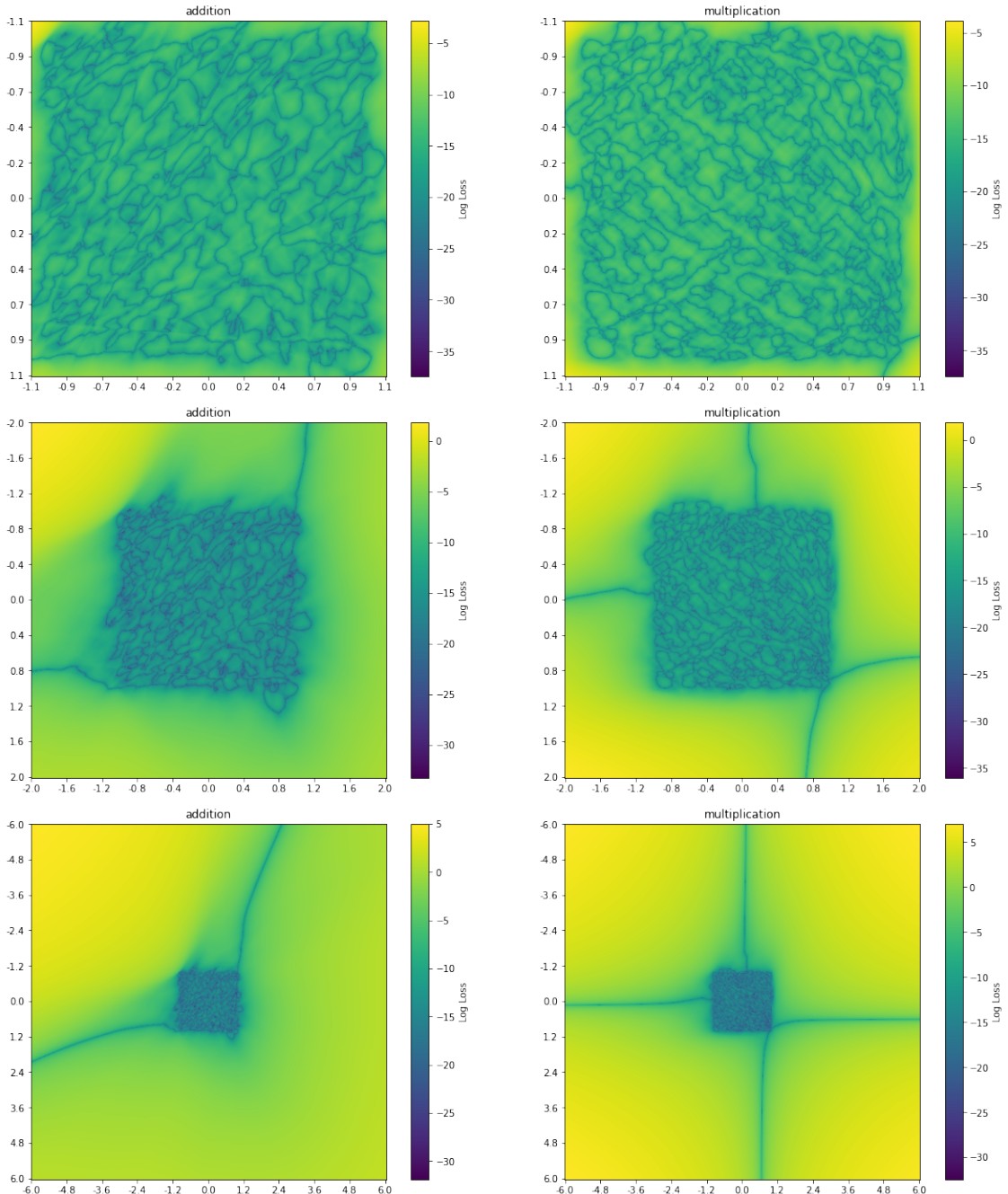

Figure 13: **Addition and Multiplication for ReLU Network**. Results of ReLU network trained on both addition (Fig. 1) and multiplication. Colorbar indicates the log of the loss function (equation 3).

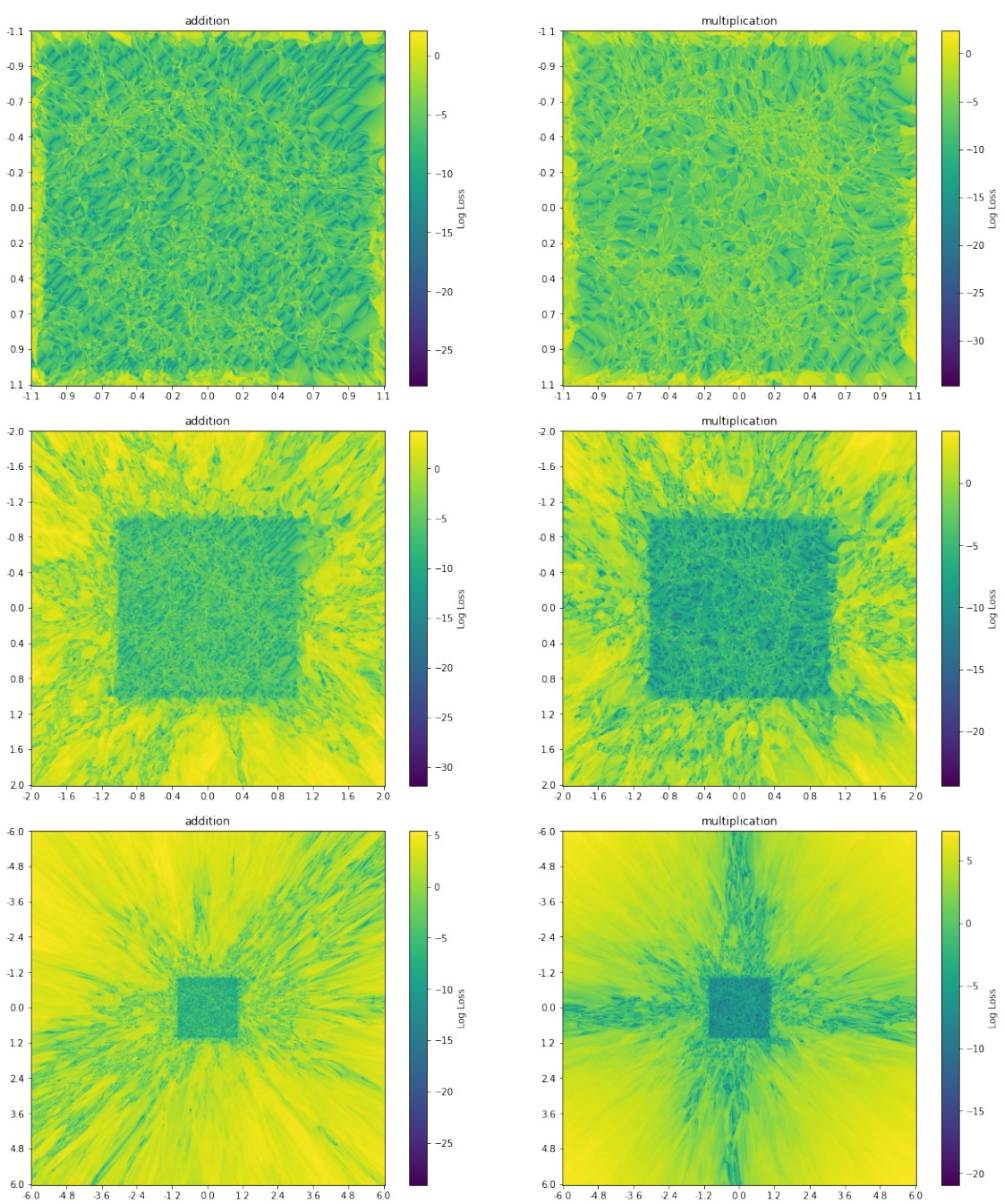

Figure 14: **Addition and Multiplication for TanH Network**. Results of TanH network, but without weight normalisation (equation 7), trained on both addition (Fig. 1) and multiplication. Colorbar indicates the log of the loss function (equation 3).

