# OpenReview forum: "Controlling Neural Network Smoothness for Neural Algorithmic Reasoning"
_TMLR — Accepted by TMLR_

### Review · Reviewer_yr94 · 2022-12-18

**Summary Of Contributions:**

This paper studies the problem of learning to add real numbers with MLPs. They find that neural networks are not able to do so in a way that generalizes, and find very rough decision boundaries as networks grow wider. They partially remedy this by making sure the functions learned by neural networks are smooth by controlling the standard deviation of the weights.

**Audience:**

Yes

**Claims And Evidence:**

Yes

**Requested Changes:**

It would be great to see:
- experiments on more difficult tasks
- discussion on normalization layers in neural networks
- discussion on how LLMs can do addition
- discussion on prior work relating smoothness and robustness

**Strengths And Weaknesses:**

Strengths:

- Overall, the idea of studying simple problems where it is well known what is contained within the networks training set seems like a very promising way to measure OOD performance. Another example of this is [1].
- The analysis leads the authors to propose and test a modification to neural networks such that the functions which they learn are more smooth. They do so by explicitly controlling the standard deviation of the weights. Experiments show that this helps learn smoother network functions, and improves robustness to adversarial attacks and noise.

Weaknesses:
- Experiments are on MNIST difficulty problems.
- No discussion on normalization layers (e.g., [2]) which are present in neural networks today and how the proposed modification relates to those. While those apply to activations instead of weights, they presumably have similar effects.
- No discussion on prior work relating smoothness and robustness (e.g., [3]).
- No discussion on that modern neural networks can do addition (e.g., [4]).

Minor:
- Eq 6 contains a typo - should be $\frac{sinh(x)}{cosh(x)}$.

[1] https://arxiv.org/abs/2107.12580
[2] https://arxiv.org/abs/1607.06450
[3] https://proceedings.neurips.cc/paper/2019/file/ae3f4c649fb55c2ee3ef4d1abdb79ce5-Paper.pdf
[4] https://arxiv.org/abs/2211.09066

---

> ### Author Response · Authors · 2023-01-05
> **Response to Review**
>
> Thank you very much for your thorough feedback. We have included your suggestions in the paper and added additional experiments. Please find below our replies to your individual comments and questions.
>
> Thank you for catching the typo in eq. 6, we have fixed that. We have also added the reference to the Pointer Value Retrieval work [1] which is very related and inspiring for future research in similar directions.
>
> ‘experiments on more difficult tasks’ – Thank you for this suggestion, we have added experiments on CIFAR10 with the same architecture in the main paper, and with a ResNet18 convolutional neural network architecture in the Appendix. The former results mirror those on MNIST while the latter show only a marginal trend in the same direction. Please note that, as discussed in the limitations: ‘it is open for future work to investigate how the composite function of more than two layers contributes to global smoothness of a NN’.
>
> ‘No discussion on normalization layers [...] they presumably have similar effects’ – We follow Neal’s (1996) construction in normalizing the weights to approximate the NN-GP equivalence. Could you explain your intuition why normalizing the activations instead of the weights would have similar effects?
>
> ‘prior work relating smoothness and robustness (e.g., [3])’ – Thank you for this pointer, we have added the reference to the discussion, along with the reference to similar work by Fazlyab et al. (2019).
>
> ‘discussion on that modern neural networks can do addition (e.g., [4]).’ – Please note that this is the same reference as Zhou et al. (2022), which we cite and discuss throughout the paper. In particular, the last paragraph of the background section argues why we focus on MLPs instead. Please let us know if you would like us to place more emphasis on these arguments to delineate our work from attention based models.

---

> > ### Comment · Reviewer_yr94 · 2023-01-06
> > **Thanks for the response**
> >
> > With regards to normalization layers and smoothness, the hypothesis of e.g., [1] is that the main function of batch normalization is to make the optimization landscape smoother. While this is not the same as smoothing the function itself, this was the reason for my hypothesis that normalization layers would have the same effect.
> >
> > [1] https://arxiv.org/abs/1805.11604

---

> > > ### Author Response · Authors · 2023-01-11
> > > **Added discussion**
> > >
> > > Thank you so much, that is a very interesting paper and it does explain/motivate your hypothesis. We have added a paragraph on activation normalization to the discussion (updated in paper), showing different effects with regards to robustness and distribution shifts:
> > >
> > > "Here, we have closely followed the construction in Neal (1996) by normalising the network weights; future
> > > research might investigate the effect of normalising the network activations on the phenomena studied in
> > > this work. Prior findings suggests that batch normalisation (BN) (Ioffe & Szegedy, 2015), one of the most
> > > commonly used activation normalisation schemes (Ren et al., 2016), helps network training and generalisation
> > > by making the loss landscape smoother and reducing the norm of the weight gradient during optimisation
> > > (Santurkar et al., 2018). However, despite this increased smoothness of the loss landscape, BN does actually
> > > seem to hurt adversarial robustness (Benz et al., 2021; Singla et al., 2021) and make adversarial training
> > > more difficult (Wang et al., 2022; Walter et al., 2022). By contrast, test-time BN can help models deal with
> > > distribution shifts (Schneider et al., 2020), and normalisation in the form of feature competition (e.g., sparse
> > > coding, divisive normalisation) has been shown to increase adversarial robustness (Paiton et al., 2020)."
> > >
> > > We thank you kindly for the pointer that prompted this text addition. We hope that this will encourage future research into these questions!

---

### Review · Reviewer_oixe · 2022-12-22

**Summary Of Contributions:**

This work presents an empirical study on how a few design choices affect the performance of a two-layer neural network (NN) when trained to approximate the addition of two real numbers, i.e., ($f(x_1, x_2)=x_1+x_2$). The key result is a visualization of how the error surface of the NN changes depending on the level of overparameterization (i.e., width). The authors visually compare these plots with the error surfaces obtained by training Gaussian proceesses (GPs) with varying degrees of smoothness on the same task. They highlight the qualitative similarity between the complex error surfaces of the GPs with low smoothness and the heavily overparameterized NNS. As higher degrees of smoothness for GPs lead to better performance on the addition task, the authors borrow inspiration from the equivalence results between GPs and NNs in certain settings (e.g., infinfinite-width, certain initializations, and activation functions) and propose a simple wight normalization strategy which they claim can control the smoothness of a certain small class of NNs. The authors support this claim showing better qualitative results and a better visual correspondence between GPs and the NN in a few settings. Finally, the paper presents preliminary results on MNIST in which the proposed weight normalization strategy applied to a two-layer NN slightly improves the robustness of the NN to additive Gaussian perturbations and FGSM adversarial attacks.

Overall the main contributions of this work are:
1. A visual comparison between the error surfaces of two-layer NNs and GPs trained to approximate $f(x_1, x_2)=x_1, x_2$ varying the network width and the GP smoothness.
2. A quantitative analysis of the generalization error in and out of the support of the input distribution when this support is a narrow ring. GPs generalize better on this ring, but also inside. Neither the NNs nor the GPs extrapolate outside the ring.
3. A few experimental results showing that a GP-inspired weight normalization strategy can control the smoothness of a two-layer NN in a one-dimensional sinusoidal regression task, and the previously described two-dimensional addition task.
4. Preliminary results on MNIST showing a slight increase in robustness to additive noise and weak adversarial attacks using this weight normalization strategy.

**Audience:**

No

**Broader Impact Concerns:**

I see no clear broad impact concerns stemming from this theoretical work.

**Claims And Evidence:**

No

**Requested Changes:**

Unfortunately, I believe a lot of work would be required for this manuscript to meet the bar of acceptance to TMLR. In particular, I think it is critical that the presented results go beyond the shallow anecdotal evidence they currently provide, and give new insights which add value on top of the prior literature. This could be done by thoroughly studying theoretically the simple setting described in this work (note that there exist many mathematical tools today to study two-layer MLPs on simple tasks, e.g., NTK limit, mean-field, etc), or scaling up the experiments to more interesting settings from a relevant practice, isolating new phenomena that could inform future practice or theory on deep learning. I personally believe such studies would be beyond the scope of what is possible in a major revision of this work.

**Strengths And Weaknesses:**

## Strengths

1. **Relevant research question**: Understanding the inductive bias of neural networks when solving algorithmic tasks is a topic of great interest for the ML/DL research community.
2. **Educative illustration of error surface of NNs and GPs**: Personally, I find the collection of figures illustrating the error surface of the different NNs and GPs quite interesting and pedagogical. They clearly show what is the role of smoothness of the GP prior, how the representation of the NNs and GPs adapts to the training support, and how both hypotheses spaces favor solutions with connected paths of low error in the input space.
3. **Clear writing**: Overall, this manuscript is very well-written and it is very easy to read.

## Weaknesses

1. **Anecdotal empirical evidence on extremely toy settings**:  In their introduction, the authors claim that the main weakness of the prior work on this topic was that "...it provided solutions to a specific problem without any transferable insights into neural network functions as we obtain in this paper". However, this work only provides a few empirical results on an extremely toy setting (e.g., two-layer neural networks on a single two-dimensional linear problem) that vastly differs from real practice. In fact, in Section 4, the authors directly claim as their main limitation that "the proposed modifications are only applicale to simple two layer MLPs".
2. **Shallow observations with little added value over deeper prior work**: The previous weakness would not be major if the performed analysis provided a deep understanding of new phenomena. However, the observations described in this paper are only superficial and provide little additional insights over prior work:
	- The connections between GPs and NNs have been widely studied in previous literature (far beyond Neal (1996)).
	- There exist deep empirical and theoretical studies analyzing the extrapolation abilities of wide MLPs with far stronger insights than the ones described in this work, e.g., (Xu et al. 2021).
	- The toy sinusoidal regression studied in Section 3.3. has been a deep object of study of many works on the spectral bias of NNs, e.g., (Rahaman et al. 2019).
3. **Lack of rigor in robustness study**: The connections between function smoothness and robustness are widely known in the robustness community and have been studied to a much greater extent than what is presented in this study. The minimal improvements in robustnesss presented in this work for MNIST on a trained MLP and evaluated using FGSM are not relevant for a community which in 2022, after having shown several times that weak robustness evaluations in toy settings did not provide meaningful research signals, has developed strong standardized benchmarks and codes of good practices to avoid pitfalls in robustness evaluations due to lack of rigor in research (Croce et al. 2021).
4. **Inconsistent experimental settings**: I find intriguing that despite the simplicity of the problem under study, depending on the section, some experiments are arbitrarily performed using different sets of parameters. For example, why are the experiments on the ring performed with $N\to\infty$ , but the experiments on the disk for $N=256$? Similarly, some figures have colormaps and some others do not, and while the first half of the paper deals with ReLU networks, the second half uses $\tanh$ and weight normalization.

- Xu et al. "How neural networks extrapolate:  from feedforward to graph neural networks". ICLR 2021
- Rahaman et al. "On the spectral bias of neural networks". ICML 2019.
- Croce et all. "RobustBench: a standardized adversarial robustness benchmark". NeurIPS 2021

---

> ### Author Response · Authors · 2023-01-06
> **Response to Review**
>
> Thank you very much for your thorough feedback. We much appreciate that you find our work ‘relevant’, ‘well written’ and ‘pedagogical’, this is important for us since we hope that an engaging presentation of these new phenomena (i.e., ‘how the error surface of the NN changes depending on the level of overparameterization’) will inspire further research. We have included your suggestions in the paper and added additional experiments. Please find below our replies to your individual comments and questions.
>
> ‘In their introduction, the authors claim that the main weakness of the prior work on this topic was that "...it provided solutions to a specific problem without any transferable insights into neural network functions as we obtain in this paper"’ – Thank you for pointing this out! There was, indeed, a ‘iii)’ missing in the last paragraph of the Background section. The critique that you quoted does only apply to i) and ii) (i.e., handcrafted solution in our MLP setting) and not to the attention based work in iii). We hope that this is clear now.
>
> ‘little added value over deeper prior work’ – As you stated in your review, our key contribution in this paper is to expose how MLPs fails on this simple task, learning an intricate error surface. Regarding the prior work that you refer to: We build on the work by Neal (1996), but we do not aim to extend it beyond making the variance/smoothness link more explicit; The Xu et al. [2021] is very interesting, especially, because their theorem 2. is at odds with our demonstration how MLPs fail to learn addition (i.e., a linear function) – ideally, this might trigger further theory about MLP behavior in the non-asymptotic (finite width and data) regime; Analogously, the Rahaman et al. (2019) reference is interesting because it suggests a low-frequency bias of the functions learned by MLPs, which makes the rough (high frequency) error surfaces that we find even more surprising. Judging the theoretical depth of these prior works is very subjective. However, we would like to state again that our findings are, at least, surprising given this prior theory and should thus be evaluated on the merrit of exposing a new phenomenon as fertile inspiration for future work. We have added both references to our discussion, thank your for the pointers!
>
> ‘Lack of rigor in robustness study’ – Thank you for pointing this out, we fully agree that adversarial robustness research has suffered from weak evaluations (see also Tramer, Carlini, Brendel, Madry, 2020, NeurIPS). We had initially only included a simple FGSM attack to demonstrate that our smoothness arguments transfer to a simple MNIST example, which was intended as a little tangent as the last section of our paper. But you are absolutely right in demanding that any robustness claims need to be properly vested. Therefore, we now include an additional experiment where we use the stronger AutoAttack (Croce et al., 2020, ICML), see Appendix A.2.3 Fig. 8. We observe the same qualitative results and the same ordering among our models. This confirms the findings we obtained previously with the faster FGSM attack.
>
> ‘Inconsistent experimental settings’ – Thank you for paying attention to these details, which we have chosen very much on purpose: As we argue in section 3.2. ‘to ensure that the findings from fig. 1 do not depend on limited training data or finite network size, we set the number of hidden units to N=10,000 and generate a new random batch for every gradient step’. Fig. 2 has colormaps because it explicitly aims at quantifying generalization performance. Figs. 1 and 4, by contrast, expose qualitative results about the shape of the model’s loss surface. The switch to weight normalization and TanH nonlinearities is key in following Neal’s (1996) construction, we have changed the text in section 3.3. to give more emphasis to this important detail. Thank you again for pointing out those ambiguities.
>
> Overall, we are under the impression that you would have liked to read a different paper with other approaches to the presented problem. We found the equivalence between NNs and GPs very instructive in this present work. However, we emphasize that our focus in this present work is on the clear/pedagogical exposition of a new phenomenon to inspire future research, in your words: ‘isolating new phenomena that could inform future practice or theory on deep learning’. We repeat that prior work did not predict these loss surfaces and even makes them surprising, if not at odds with their theory. We very much look forward to future theoretical developments that might explain these observations with the favorite analysis tools (e.g., NTK, mean field etc.) of different researchers.

---

> > ### Comment · Reviewer_oixe · 2023-01-11
> > **Thanks for the response**
> >
> > Thank you very much for your thoughtful response, and please let me first apologize if the tone of my previous review sounded too harsh, unpolite or personal. I appreciate the effort of the authors in writing this paper and my comments purely concern the correctness of the technical content in this manuscript. Let me also thank you for taking my comments and the suggestions of the other reviewers into account when improving your work.
> >
> > There are two main criteria for acceptance to TMLR:
> >
> > > [TMLR Criterion #1] Are the claims made in the submission supported by accurate, convincing and clear evidence?
> >
> > > [TMLR Criterion #2] Would at least some individuals in TMLR's audience be interested in knowing the findings of this paper?
> >
> > After having read the new version of the manuscript and the comments of the other reviewers, I believe that this work technically meets  Criterion #1 (i.e., the results are factually correct) but does not meet Criterion #2 (i.e., it adds little value to the community).
> >
> > As far as I know the results showing that these particular MLPs in these narrow settings cannot learn one-dimensional addition is novel and probably correct, but in my opinion the authors have only superficially studied it. Most results are anecdotal and lack the depth or breadth to appeal what I believe is the TMLR audience interested in deep learning theory. Similarly, the results on adversarial robustness might be technically correct (thanks for running them on CIFAR10 and evaluating with AutoAttack), but are of little interest to adversarial robustness experts as they cannot be generalized to standard architectures and provide only marginal improvements.
> >
> > I agree that deciding what interests the TMLR audience is a subjective guess which makes assessing Criterion #2 much harder (thus the need to have multiple reviewers).  I will recommend rejection based on this, but I would understand if the AE and the other reviewers had different opinions about it and considered that "in-my-opinion" these toy results qualify for acceptance to TMLR.

---

> > > ### Author Response · Authors · 2023-01-12
> > > **Thank you**
> > >
> > > Thank you very much for being so conciliatory. The review process can be harsh but we do really appreciate your questions and thoughts. They have triggered a number of future analysis ideas that we are very keen to discuss and investigate in the next research project. In any case, we hope that you have gained at least some insights and inspirations from reading our work. Your feedback and inputs have certainly helped to improve its rigor and quality, so thanks again for your time and consideration!

---

### Review · Reviewer_ksSX · 2022-12-22

**Summary Of Contributions:**

This submission seeks to understand the properties of multi layer perceptrons when solving the task of adding real numbers. They investigate generalization and extrapolation properties, contrasting them with those of gaussian processes. Motivated by GPs, they propose to regularize the variance of the weights at initialization and throughout training, which helps improve the functional smoothness.

**Audience:**

Yes

**Broader Impact Concerns:**

No broader impact concerns

**Claims And Evidence:**

Yes

**Requested Changes:**

I would recommend acceptance if the authors conduct a more thorough investigation of why neural networks are not able to solve the addition task. I am not sure how exactly this could be done, but listed some potential ideas above in "Weaknesses." Perhaps it is useful to understand how the behavior in function space evolves throughout training.

It would also be nice to have experiments with convolutional neural networks or attention-based networks (which have been shown to perform addition by Zhou et al.), which would also potentially help answer the questions of what ails simple MLPs, since we know more complicated architectures can solve the task. This would also show whether benefits to regularizing the variance scale to tasks and architectures more relevant to practitioners.

**Strengths And Weaknesses:**

## Strengths
- This submission motivates the need to investigate addition through the lens of neural algorithmic reasoning. The motivation for investigation on a simple problem is clear.
- They illustrate that a simple two layer MLP is not capable of learning the addition task.
- The authors connect their work to Gaussian processes and motivate a length scale for neural networks, which show benefits to smoothing the learned function and in adversarial robustness.

## Weaknesses
- An important scientific question that feels unsatisfactorily addressed after reading the paper is an explanation of why existing neural networks fail to generalize to solve addition, given that many such solutions exist in a subspace. It seems important to diagnose which part of the training algorithm fails (e.g. is it due to network initialization, learning rate, how data is generated, how the data is batched, or whether we use SGD or Adam?). A clear diagnosis for why simple MLPs fail could also be useful if it illustrated a more general failure cause of neural networks.
- It's not clear how to set the variance hyperparameter, beyond retraining the network multiple times. Moreover, it seems important to justify the choice of scaling weights throughout training by comparing to something like weight decay.
- Some experimental design choices need to be better motivated. The initial motivation studies the ReLU activation, but the authors later switch to the tanh activation. It's not clear from the existing experiments how much the choice of number of units matters e.g. what happens when N > 10000?

---

> ### Author Response · Authors · 2023-01-05
> **Response to Review**
>
> Thank you very much for your thorough feedback. We have included your suggestions in the paper and added additional experiments. Please find below our replies to your individual comments and questions.
>
> ‘why existing neural networks fail to generalize to solve addition, given that many such solutions exist in a subspace’ – To address this question we have performed an additional experiment where we observe the loss of the model as we interpolated between the weights that are learned during training \theta_{mse} and those from the optimal solution \theta_{opt} (eq. 1). The results are shown here {https://ibb.co/BtQNq5n}. We can see that the optimal solution is the global minimum (up to machine precision), however, it is a substantially sharper minimum than the one found by gradient descent. Thus, it is much harder for the model to converge to this global minimum.
>
> ‘how to set the variance hyperparameter’ – Yes, this hyperparameter has to be chosen in practice depending on the requirements (see, e.g., the performance robustness tradeoff in Fig. 5). However, we see it as a future avenue to use the theoretical results from Lederer et a. (2019) to directly specify \sigma as a function of the desired Lipschitz constant.
>
> ‘it seems important to justify the choice of scaling weights throughout training by comparing to something like weight decay.’ – We agree, which is why we included this as a control in Appendix 2.2. Fig. 7.
>
> ‘Some experimental design choices need to be better motivated. The initial motivation studies the ReLU activation, but the authors later switch to the tanh activation. It's not clear from the existing experiments how much the choice of number of units matters e.g. what happens when N > 10000’ – Thank you for pointing this out, we have further highlighted the design choices in the paper (section 3.3) which, crucially (following the construction in Neal, 1996), include a TanH nonlinearity. The largest setting with N=10,000 units is also in alignment with Neal (1996) and it was the largest setting that did not result in OOM errors.
>
> ‘It would also be nice to have experiments with convolutional neural networks or attention-based networks’ – Thank you for this suggestion, we have added experiments on CIFAR10 with the same architecture in the main paper, and with a ResNet18 convolutional neural network architecture in the Appendix. The former results mirror those on MNIST while the latter show only a marginal trend in the same direction. Please note that, as discussed in the limitations: ‘it is open for future work to investigate how the composite function of more than two layers contributes to global smoothness of a NN’. Regarding attention-based networks, we appreciate your comment: ‘The motivation for investigation on a simple problem is clear’ and will try to clarify it even more in the background section that in this work we focus on MLPs as the simplest and most ubiquitous representative of neural networks.

---

> > ### Comment · Reviewer_ksSX · 2023-01-13
> > **Thanks for the response**
> >
> > Thanks, I appreciate the response. I think the proposed experiment is a good first step in answering the question "why existing neural networks fail to generalize to solve addition, given that such solutions exist in a subspace" The attached image you provided does not load for me, and I think describing and analyzing the results of the experiment would be useful (perhaps in the appendix).
> >
> > From figure 7, it seems to me like the results of rescaling the weights compares similarly to weight decay?
> >
> > I also can't seem to find to find results of the ResNet experiment in the paper.

---

> > > ### Author Response · Authors · 2023-01-18
> > > **Experiments and Details added to Appendix**
> > >
> > > Thank you so much! The link does seem to be broken and we agree that those interpolation results should be in the paper. We have added them as a new section (A.2.2) to the Appendix along with two tentative explanations. We have also added a closer comparison between the TanH+weight_normalisation and ReLU+weight_decay to the Appendix (A.2.5). We see that the gain in accuracy is larger for the TanH model on adversarial robustness, but slightly smaller on noise robustness. We have adjusted the main text accordingly. Finally, we have also added the ResNet experiments and details to the Appendix and apologize (these updates should have already been included in the previous revision). Moreover, we have made a correction, namely we used a ResNet9 (not ResNet18). This gives support to the negative results that even comparatively shallow multilayer models require further research to construct a (bounded) length-scale of the composite kernel (possibly following Lee et al., 2017).

---

> > > > ### Comment · Reviewer_ksSX · 2023-01-20
> > > > **Looks good**
> > > >
> > > > For the interpolation experiment, I would recommend running the experiment with multiple seeds and taking a look at "What can linear interpolation of neural network loss landscapes tell us?" (Vlaar and Frankle) and "Analyzing Monotonic Linear Interpolation in Neural Network Loss Landscapes" (Lucas et al.) to see if there is analysis to strengthen the claim. I will update my "Claims And Evidence" score.

---

> > > > > ### Author Response · Authors · 2023-01-23
> > > > > **More experiments =)**
> > > > >
> > > > > Thanks again, those references are very interesting and they have inspired a few extensions to our interpolation experiments (revision uploaded). Briefly, we show evidence that: i) greedy gradient descent might prefer a sub-optimal solution; ii) most training progress can be recovered by optimizing the second layer (read out) weights, which alone is not enough to reach the optimal solution; and iii) the quality of the first layer feature space is nearly constant in parameter space, except for a sharp peak around the optimal solution, which makes it very hard to find during gradient descent optimization. For these experiments, we have added discussion of the following references:
> > > > >
> > > > > James Lucas, Juhan Bae, Michael R Zhang, Stanislav Fort, Richard Zemel, and Roger Grosse. Analyzing monotonic linear interpolation in neural network loss landscapes. arXiv preprint arXiv:2104.11044, 2021.
> > > > >
> > > > > Tiffany J Vlaar and Jonathan Frankle. What can linear interpolation of neural network loss landscapes tell
> > > > > us? In International Conference on Machine Learning, pp. 22325–22341. PMLR, 2022.
> > > > >
> > > > > Rahim Entezari, Hanie Sedghi, Olga Saukh, and Behnam Neyshabur. The role of permutation invariance in
> > > > > linear mode connectivity of neural networks. arXiv preprint arXiv:2110.06296, 2021.
> > > > >
> > > > > Samuel K Ainsworth, Jonathan Hayase, and Siddhartha Srinivasa. Git re-basin: Merging models modulo permutation symmetries. arXiv preprint arXiv:2209.04836, 2022.
> > > > >
> > > > > Keller Jordan, Hanie Sedghi, Olga Saukh, Rahim Entezari, and Behnam Neyshabur. Repair: Renormalizing
> > > > > permuted activations for interpolation repair. arXiv preprint arXiv:2211.08403, 2022.
> > > > >
> > > > > Arthur Jacot, Franck Gabriel, and Clément Hongler. Neural tangent kernel: Convergence and generalization
> > > > > in neural networks. Advances in neural information processing systems, 31, 2018.
> > > > >
> > > > > Thank you again for suggesting these investigations!  We have found those new results very insightful (also more broadly in the context of the new references above and recent research on model interpolations) and hope that they have added value to our paper, as we now have a better understanding why model fails to learn the optimal solution. Please let us know if anything is unclear or if you have further questions!

---

> > > > > > ### Comment · Reviewer_oixe · 2023-01-24
> > > > > > **A solid addition**
> > > > > >
> > > > > > Thank you very much for adding this experiment. It certainly alleviates some of my concerns regarding the possible interest of this paper by the TMLR audience (Criterion #2). In this regard, although I believe most of the robustness contributions of this paper are of little interest, I do see this experiment as a nice contribution which sheds new light on why MLPs cannot learn addition.
> > > > > >
> > > > > > Personally, I believe the paper would be stronger if it focused even more on these type of experiments and results, giving further ablations with respect to model width, depth, and so on, but the current version is enough to weakly meet the bar of acceptance in my opinion. I will change my recommendation to reflect this, and I also thank Reviewer ksSX for the nice suggestions.

---

### Decision · Action_Editors · 2023-01-26

**Recommendation:** Accept as is

**Comment:**

The claims of the paper could have been more significant if authors would try a larger range of architectures and normalization techniques. Authors could have also covered a few other algorithmic/arithmetic tasks. Nevertheless, I think the current limited setting is still interesting. I think perhaps understanding the relationship between this suggestion and known normalization techniques is the most important part that could be added to the paper.

**Audience:**

Even though the main studied task is simple, I believe studying this task and the comparisons with GP provides insights that are valuable to  some individuals in TMLR's audience.

**Claims And Evidence:**

The main focus of the paper is the task of adding two real numbers. Authors provide accurate and convincing empirical results to show the failure of some neural nets in learning such a task.

---

> ### Author Response · Authors · 2023-02-01
> **Thank you for the Reviews!**
>
> We would like to thank all reviewers and the action editor again for this review process! The feedback and questions were very thoughtful and have triggered a number of additional discussions and experiments. Overall, we believe that this has improved the quality of our paper, linking it better to prior work and motivating us to dig deeper into some aspects of the phenomenon (e.g., Appendix A.2.2). This has been an enriching experience and we hope that the reviewers have gained some value or inspiration from the time they invested into this process. Thank you very much!